# Covalent TCR-peptide-MHC interactions induce T cell activation and redirect T cell fate in the thymus

Christopher Szeto [1,2,5], Pirooz Zareie [1,5], Rushika C. Wirasinha [1], Justin B. Zhang [1], Andrea T. Nguyen [1,2], Alan Riboldi-Tunnicliffe [3], Nicole L. La Gruta [1,6] ✉, Stephanie Gras [1,2,6] ✉ & Stephen R. Daley [1,4,6] ✉

Interactions between a T cell receptor (TCR) and a peptide-major histocompatibility complex (pMHC) ligand are typically mediated by noncovalent bonds. By studying T cells expressing natural or engineered TCRs, here we describe covalent TCR-pMHC interactions that involve a cysteine-cysteine disulfide bond between the TCR and the peptide. By introducing cysteines into a known TCR-pMHC combination, we demonstrate that disulfide bond formation does not require structural rearrangement of the TCR or the peptide. We further show these disulfide bonds still form even when the initial affinity of the TCR-pMHC interaction is low. Accordingly, TCR-peptide disulfide bonds facilitate T cell activation by pMHC ligands with a wide spectrum of affinities for the TCR. Physiologically, this mechanism induces strong Zap70-dependent TCR signaling, which triggers T cell deletion or agonist selection in the thymus cortex. Covalent TCR-pMHC interactions may thus underlie a physiological T cell activation mechanism that has applications in basic immunology and potentially in immunotherapy.

The development and function of T cells depend on signals transmitted by the TCR when it engages pMHC ligands on other cells. T cells exploit variation in TCR-pMHC interactions to differentiate in the thymus and during responses to infection[1–3]. Although the probability and rate of TCR signaling increase with TCR-pMHC interaction lifetime[1,3–7], data suggest the TCR signaling deficit associated with a shorter lifetime can be offset by a fast on-rate[8,9]. This implies that a pMHC ligand can unbind from a TCR and rebind the same or another TCR[10] so that multiple TCR-pMHC binding events effectively mimic a single, longer-lived event[11]. In the immunological synapse, TCR-pMHC complexes are subjected to forces imparted by cell migration, T cell microvillar probing of the antigen-presenting cell surface[12,13], and traction by the T cell cytoskeleton[5,14]. Application of force can accelerate rupture of TCR-pMHC complexes mediated by slip bonds and prolong those that form catch bonds[15,16]. While the dynamic nature of TCR-pMHC interactions is critical for the sensitivity and specificity of antigen recognition[17], the interplay of distinct variables is a barrier to understanding the molecular regulation of T cell activation. In theory, the stability of a covalent bond between TCR and pMHC ought to extend the lifetime of TCR-pMHC complexes, and prevent them from unbinding and rebinding, thereby decreasing the number of free variables in the immunological synapse. However, artificial cross-linking of pre-formed TCR-pMHC complexes has been found to increase[18] or abolish[19] T cell activation. Whether covalent bonds form

[1]Immunity Program and Department of Biochemistry and Molecular Biology, Biomedicine Discovery Institute, Monash University, Clayton 3800 Victoria, Australia. [2]Department of Biochemistry and Chemistry, La Trobe Institute for Molecular Science, La Trobe University, Bundoora 3086 Victoria, Australia. [3]Australian Synchrotron, Australian Nuclear Science and Technology Organisation, Clayton 3800 Victoria, Australia. [4]Centre for Immunology and Infection Control, School of Biomedical Sciences, Faculty of Health, Queensland University of Technology, Brisbane 4000 Queensland, Australia. [5]These authors contributed equally: Christopher Szeto, Pirooz Zareie. [6]These authors jointly supervised this work: Nicole L. La Gruta, Stephanie Gras, Stephen R. Daley. ✉e-mail: nicole.la.gruta@monash.edu; S.Gras@latrobe.edu.au; s5.daley@qut.edu.au

naturally between TCR and pMHC is unknown, and if so, how this would affect T cell development and activation is also unknown.

Thymic lymphocytes (thymocytes) prepare for selection by assembling genes that encode a TCRα chain and a TCRβ chain[20]. Each TCRα and TCRβ chain has three complementarity-determining region (CDR) loops, which contact pMHC antigens[21] and amino acids near the middle (apex) of CDR3 often contact the peptide[22]. An estimated ~80% of nascent TCRαβ[+] thymocytes die at the pre-selection stage in the thymic cortex due to an absence of TCR signaling[23–25]. Weakly TCR-signalled thymocytes differentiate into T-conventional (T-conv) cells, whereas strongly TCR-signalled thymocytes undergo apoptotic deletion[26]. Rare thymocytes survive strong TCR signaling and undergo "agonist selection" into specialized T cell lineages, including CD8α[+] CD8[−] intestinal intraepithelial lymphocytes (CD8αα IEL)[27,28] and CD4[+] Foxp3[+] T-regulatory (T-reg) cells[3,29]. TCRs with cysteine (Cys) within two positions of the apex of CDR3 in TCRα (CDR3α) or CDR3β are rarely expressed by CD4[+] T-conv, CD8αβ[+] T-conv and T-reg cells, whereas such TCRs are frequently expressed by CD8αα IEL and their thymic precursors[30]. This differential expression of Cys-containing CDR3 was postulated to arise from effects of disulfide (S-S) bond formation between Cys in the TCR CDR3 and Cys in pMHC self-antigens[30–32]. However, functional evidence to support this postulate has not been reported.

Here, by introducing Cys substitutions into a previously described murine TCR, we show a Cys residue at the apex of CDR3α or CDR3β is sufficient to redirect T cell fate in vivo. Furthermore, by introducing amino acid substitutions in the peptide epitope, we demonstrate S-S bond formation between a TCR and various pMHC ligands. These S-S bonds produce long-lived TCR-pMHC complexes, which can facilitate T cell activation even when the initial TCR affinity for pMHC is low.

With fewer variable parameters than noncovalent interactions, covalent TCR-pMHC interactions may be used to gain new insights into the molecular regulation of T cell activation. This physiological modality of T cell activation may also be harnessed in immunotherapy.

## Results

### T cell fate altered by a cysteine in CDR3

The 6218 TCR derives from a T cell hybridoma specific for an influenza Polymerase Acidic (PA) peptide (SSLENFRAYV) presented by H2-D[b] [33,34] and has been shown to facilitate CD8αβ[+] T-conv differentiation in C57BL/6 (B6) mice[33]. To test whether a free Cys residue in CDR3 is sufficient to affect T cell fate, we made variants of the 6218 TCR with Cys at the CDR3α or CDR3β apex; these variants were named 6218αC and 6218βC, respectively (Fig. 1a). We retrovirally transduced bone marrow (BM) cells from recombinase activating gene 1-deficient (Rag1[−/−]) mice with a construct encoding the 6218, 6218αC or 6218βC TCR plus green fluorescent protein (GFP) as a marker. To avoid complications associated with monoclonal T cell populations, we transferred equal numbers of T cell-depleted wild-type BM cells and TCR-transduced Rag1[−/−] BM cells into irradiated Rag1[−/−] recipients. In the resulting TCR-retrogenic mice[35], the group expressing the 6218 TCR had more GFP[+] CCR7[+] thymocytes than the 6218αC and 6218βC groups, and a similar trend was observed in GFP[+] CCR7[−] thymocytes (Fig. 1b and Supplementary Fig. 1), suggesting the Cys substitutions induced thymocyte deletion. The 6218 TCR induced efficient development of CD8αβ[+] T-conv cells, characterised by high expression of CCR7 and CD8β, whereas the 6218αC and 6218βC TCRs did not (Fig. 1b). In mice expressing 6218αC or 6218βC, most of the GFP[+] TCRβ[+] thymocytes had a CD4[−] CD8[−] phenotype and expressed high levels of PD-1 or NK1.1 (Fig. 1b), characteristic of IEL precursor (IELp) cells[27,36].

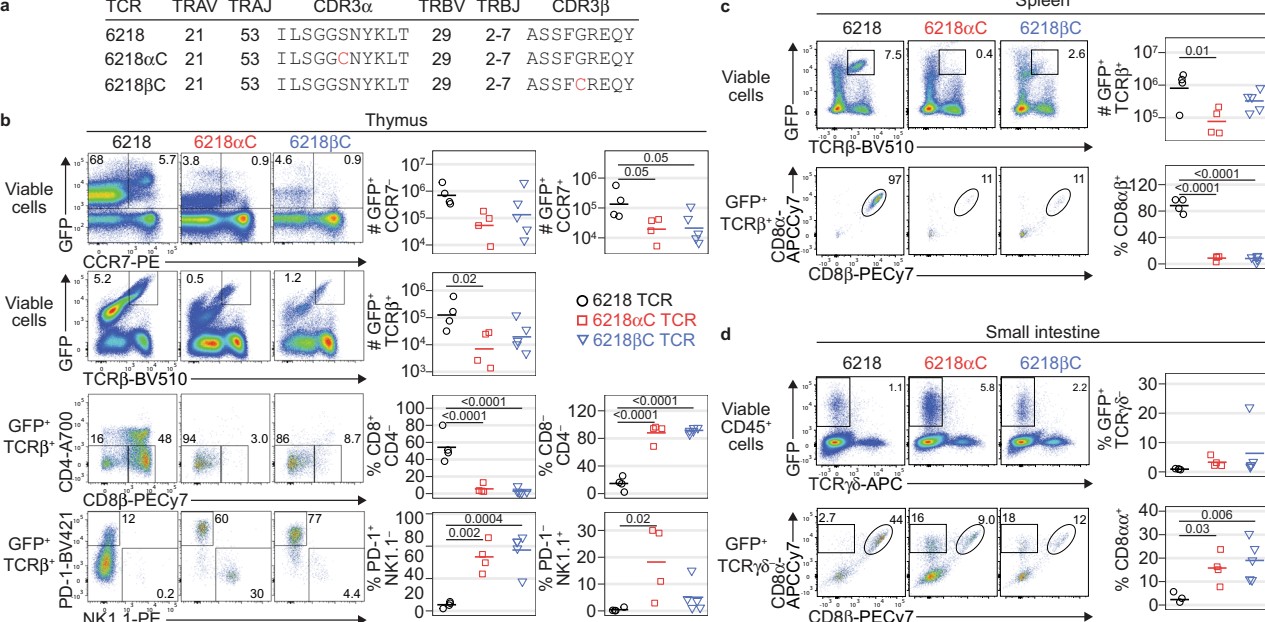

**Fig. 1 | T cell fate altered by a cysteine in CDR3. a** Details of the 6218, 6218αC and 6218βC TCRs. **b** Incorporation of Cys at the CDR3 apex alters T cell fate. BM cells pooled from Rag1[−/−] mice (7 female and 6 male; aged 37-134 d) were transduced with retroviruses encoding GFP and the 6218, 6218αC or 6218βC TCR and then mixed 1:1 with T cell-depleted BM cells from a male B6 mouse before injection into irradiated Rag1[−/−] mice (3 female and 1 male for 6218; 4 female for 6218αC and 5 male for 6218βC). The resulting TCR-retrogenic mice were analysed 5 weeks after BM transfer at 90-171 days of age. Plots show the GFP/CCR7 (row 1) and GFP/TCRβ (row 2) FACS phenotypes of live thymocytes, with a gate for GFP[+] TCRβ[+] thymocytes that were analysed for CD4/CD8β (row 3) and PD-1/NK1.1 (row 4). **c** GFP/TCRβ

phenotype of splenocytes (top row) with a gate for the GFP[+] TCRβ[+] subset, which was analysed for CD8α/CD8β phenotype (bottom row). **d** GFP/TCRγδ phenotype of small intestinal CD45[+] cells with a gate for the GFP[+] TCRγδ[−] subset that was analysed for CD8α/CD8β phenotype (bottom row). In **b**, **c**, and **d**, graphs show the absolute number (#) of gated events, determined by multiplying the percentage of gated events by the total number of cells per organ, or the percentage of gated events, as denoted on the y-axes. See Supplementary Figs. 1, 2 for gating strategies and additional data. Each symbol in a graph represents one mouse; black for 6218, red for 6218αC and blue for 6218βC. P values were determined using 1-way ANOVA with Tukey's multiple comparisons test. Source data are provided as a Source Data file.

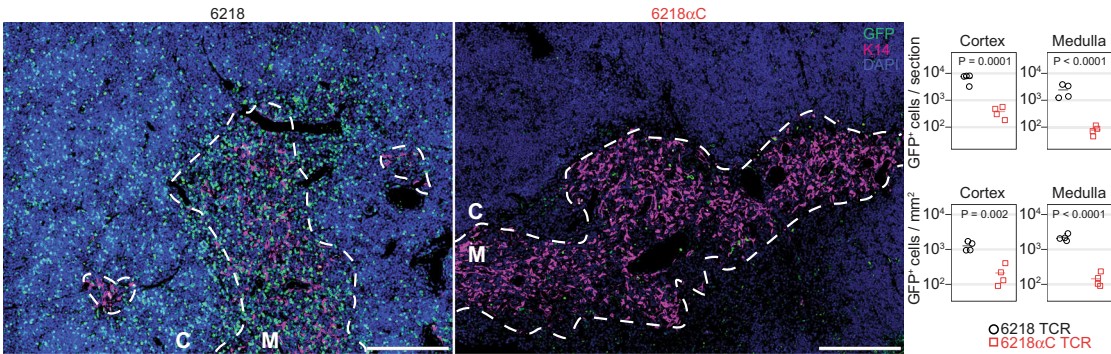

**Fig. 2 | Thymic cortical T cell deletion induced by a CDR3 cysteine.** TCR-retrogenic mice expressing the 6218 or 6218αC TCR were made as described in Fig. 1, except that all recipients were male B6 mice ($n = 4$ for 6218; $n = 4$ for 6218αC). TCR-retrogenic mice were analysed 5 weeks after BM transfer at 99-138 days of age. Thymus sections were stained for GFP (green), with medullary areas identified by staining for cytokeratin-14 (K14, magenta). Dashed lines demarcate cortex (C) from medulla (M); scale bars: 200 µm. Graphs shows the number of GFP⁺ cells per section (top) or per mm² (bottom) in cortical and medullary areas. Each symbol in a graph represents one mouse. P values were determined using unpaired two-tailed $t$ tests. Source data are provided as a Source Data file.

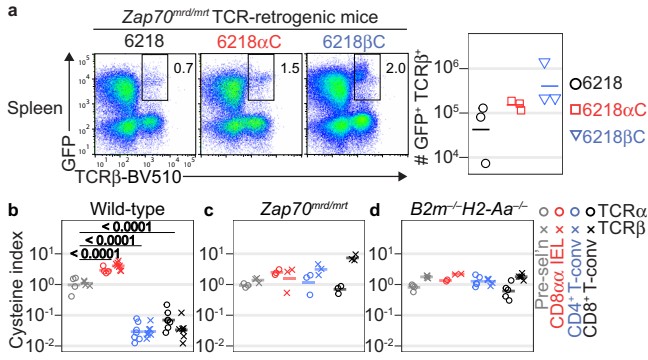

**Fig. 3 | Role of Zap70 and MHC in cysteine-linked T cell fate skewing.**
**a** Attenuation of TCR signaling disrupts the effect of a Cys-containing CDR3 on T cell fate. BM cells pooled from *Zap70^mrd/mrt* mice (11 female and 9 male; aged 33-60 d) were transduced with retroviruses encoding GFP and the 6218, 6218αC or 6218βC TCR and then injected into irradiated male *Zap70^mrd/mrt* mice. TCR-retrogenic mice were analysed on day 36 or 41 after transplantation ($n = 3$/group). Plots show splenocytes analysed for GFP/TCRβ phenotype with a gate for GFP⁺ TCRβ⁺ cells. Graph shows the absolute number of GFP⁺ TCRβ⁺ splenocytes with each symbol representing one mouse. **b–d** Differential Cys-containing CDR3 expression in polyclonal T cell subsets requires strong TCR signaling in response to pMHC ligands. Sorted pre-selection thymocytes, small intestinal CD8αα IEL, splenic CD4⁺ T-conv and splenic CD8⁺ T-conv populations were analysed by TCR sequencing. Graphs show the percentage of unique TCRα (circles) or TCRβ (crosses) sequences with Cys within 2 positions of the middle CDR3 amino acid (cysteine index) with each symbol on a graph representing a sample from one mouse. **b** Some of the data from wild-type mice has been published[30] and is included here for comparison ($n = 11$; 6 females and 5 males, aged 45-113 d); **c** Data for *Zap70^mrd/mrt* mice (n = 3 males aged 37–74 d); **d** Data for *B2m^−/−H2-Aa^−/−* mice (n = 1 female and 4 males aged 98–140 d). *P* values in **b** were determined using 2-way ANOVA with Sidak's multiple comparisons test. Source data are provided as a Source Data file.

In the spleen, GFP⁺ TCRβ⁺ CD8αβ⁺ T-conv cell numbers were highest in the 6218 group (Fig. 1c). In the small intestine, however, GFP⁺ CD8αα IEL populations were larger in the 6218αC and 6218βC groups (Fig. 1d). Similar results were obtained in a separate experiment using *Tcra^−/−* BM donors and recipients (Supplementary Fig. 2). A Cys residue at the CDR3α or CDR3β apex is thus sufficient to skew thymocyte fate away from CD8αβ⁺ T-conv cell development and towards deletion or CD8αα IEL development.

To assess where in the thymus T cell development was affected by a Cys in CDR3, we stained thymic sections with antibodies against GFP and the medullary marker cytokeratin-14 (K14). GFP⁺ cells were significantly more frequent in both the cortex and the medulla of mice expressing the 6218 TCR compared to mice expressing the 6218αC TCR (Fig. 2). The effect on GFP⁺ cell numbers in the cortex indicates that the Cys substitution triggered thymocyte deletion at an immature stage of T cell development in the thymic cortex[20].

### Role of Zap70 and MHC in cysteine-linked T cell fate skewing

In mice with attenuated function of the TCR signaling protein, Zap70, thymocytes that would normally be deleted undergo aberrant development into CD4⁺ or CD8αβ⁺ T-conv cells[37,38]. To explain that finding, TCR-pMHC interactions that should induce strong TCR signaling are thought to induce only weak TCR signaling due to the attenuated signal transmission through mutant Zap70[37,38]. Consistent with that interpretation, we found the percentage of nascent thymocytes that receives any TCR signal is reduced in *Zap70^mrd/mrt* mice[38], with a disproportionately large reduction in thymocytes that receive a strong TCR signal (Supplementary Fig. 3). To test whether a Cys-containing CDR3 normally elicits strong TCR signaling in vivo, we looked for evidence that thymocytes with a Cys-containing CDR3 undergo aberrant development into T-conv cells in *Zap70^mrd/mrt* mice. We analysed TCR-retrogenic mice bearing *Zap70^mrd/mrt* BM cells transduced with the 6218, 6218αC or 6218βC TCR. The numbers of GFP⁺ TCRβ⁺ cells in the spleen were similar, albeit relatively low, when these TCRs were expressed in *Zap70^mrd/mrt* cells (Fig. 3a). This result supports our hypothesis that a Cys-containing CDR3 normally elicits strong TCR signaling in vivo.

Although the TCR-retrogenic experiments above provide functional evidence that a Cys in CDR3 skews T cell fate, those experiments were limited to monoclonal T cell populations. To generalize those findings, we sequenced the TCR repertoires of polyclonal T cell populations, including in mice with genetic defects in TCR-pMHC signaling. In wild-type mice, Cys-containing CDR3 were enriched in CD8αα IEL and depleted in CD4⁺ and CD8⁺ T-conv cells compared to pre-selection thymocytes (Fig. 3b and Supplementary Fig. 4)[30]. However, pre-selection thymocytes and mature T cell subsets in *Zap70^mrd/mrt* mice had similar frequencies of Cys-containing CDR3 (Fig. 3c), indicating that polyclonal thymocytes with a Cys-containing CDR3 undergo aberrant development into T-conv cells in *Zap70^mrd/mrt* mice. To test whether the effect of a Cys-containing CDR3 depends on pMHC ligands, we sequenced the TCR repertoires of *B2m^−/−H2-Aa^−/−* mice, which lack cell-surface expression of MHC proteins[39,40]. Consistent with prior data[31], we found similar frequencies of Cys-containing CDR3 in pre-selection thymocytes and mature T cells from *B2m^−/−H2-Aa^−/−* mice (Fig. 3d). Collectively, these results demonstrate the differential expression of Cys in mature TCR repertoires is a direct consequence of

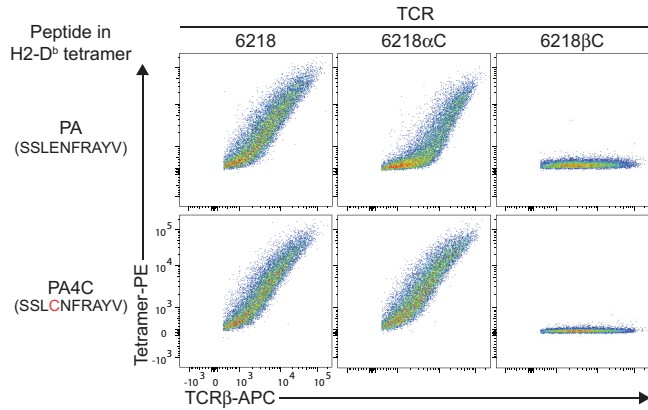

**Fig. 4 | Context-dependent effects of Cys on TCR-pMHC binding.** TCR transfectants expressing mouse CD3, GFP and the 6218, 6218αC or 6218βC TCR were incubated with anti-TCRβ and tetramers of H2-D$^b$ presenting the indicated peptide (left). FACS plots show tetramer staining versus TCRβ expression on live GFP$^+$ TCRβ$^+$ cells. Data are representative of 6 experiments for cells expressing the 6218 TCR or the 6218αC TCR and 1 experiment for cells expressing the 6218βC TCR.

Cys-containing CDR3 inducing strong TCR signaling in response to pMHC ligands in the thymic cortex.

TCR-sequencing data suggest 6-10% of CD8αα IEL and their thymic precursors have a Cys-containing CDR3[30]. To investigate whether the *Zap70*$^{mrd/mrt}$ genotype causes the broader CD8αα IEL-associated TCR repertoire to infiltrate alternative T cell lineages, we introduced the *Zap70* mutations into Yae62 TCRβ chain transgenic (Yae62β-tg) mice[41]. In Yae62β-tg mice, the expression of a given endogenous TCRα sequence is reproducibly associated with a given T cell fate[3,30]. CD4$^+$ T-conv, CD8$^+$ T-conv, and T-reg cells in *Zap70*$^{mrd/mrt}$ Yae62β-tg mice all expressed TCRα sequences that were preferentially expressed by CD8αα IEL and their thymic precursors in *Zap70*$^{+/+}$ Yae62β-tg mice (Supplementary Figs. 5, 6). The presence of a free Cys near the CDR3 apex marks a subset of TCRs within a broader TCR population that normally induces cortical deletion or CD8αα IEL development but can infiltrate alternative T cell lineages when the thymocytes carry mutations that attenuate TCR signal transmission.

## TCR-peptide S-S bonding without structural rearrangement

Next, we investigated whether a S-S bond can form between the TCR and peptide by introducing Cys substitutions into a model TCR-pMHC combination. Based on the published 6218 TCR-PA/H2-D$^b$ structure[33], we predicted the 6218αC TCR would form a S-S bond with a variant of the PA peptide containing Cys at position 4 (PA4C). Tetramers of H2-D$^b$ loaded with PA or PA4C bound to cells expressing the 6218 or 6218αC TCR, but not to cells expressing the 6218βC TCR (Fig. 4). The absence of binding to the 6218βC TCR was expected because the Gly to Cys substitution introduces a larger side-chain, which would likely alter the conformation of the CDR3β loop and prevent close interaction between its main chain and the Arg at position (P)7 of the peptide[33]. To assess effects of these Cys substitutions on the TCR-pMHC interface, we solved four high-resolution crystal structures (Supplementary Table 1). The PA4C/H2-D$^b$ structure confirmed the Cys at P4 of the PA4C peptide is exposed for potential TCR binding and the overall structure is similar to PA/H-2D$^{b\ 42}$, with a root mean square deviation on the MHC cleft of 0.34 Å and on the peptide of 0.21 Å (Supplementary Fig. 7a-c). Overlay of the 6218 TCR-PA/H2-D$^b$, 6218 TCR-PA4C/H2-D$^b$ and 6218αC TCR-PA4C/H2-D$^b$ complexes revealed similar binding modes (Fig. 5a-c, Supplementary Fig. 7d-i). We observed a S-S bond between the free Cys of the 6218αC TCR and the P4-Cys of the PA4C peptide (Fig. 5d). The S-S bond is formed without requirement for structural rearrangement of the peptide, the CDR3, or docking topology of TCR chains (Fig. 5a-e).

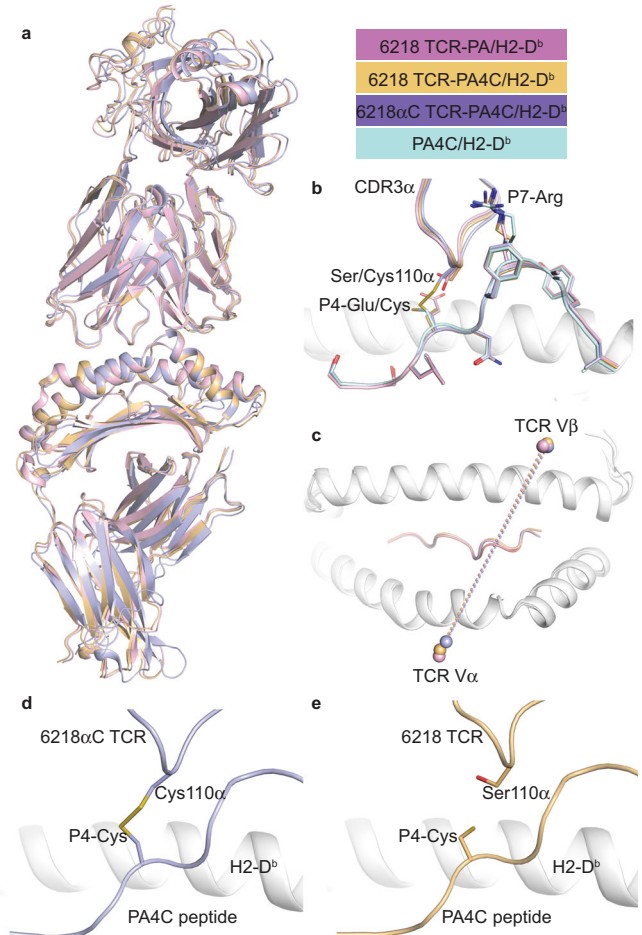

**Fig. 5 | TCR-peptide S-S bonding without structural rearrangement.**
**a** Superposition of the three TCR-pMHC structures solved in this study with the 6218 TCR-PA/H2-D$^b$ complex in pink, the 6218 TCR-PA4C/H2-D$^b$ in gold, and the 6218αC TCR-PA4C/H2-D$^b$ in purple. **b** Zoom-in view on the TCR-peptide interface in **a** with the addition of the PA4C/H2-D$^b$ structure (PA4C peptide in pale blue). **c** Top view of the H2-D$^b$ antigen-binding cleft in white cartoon with the mass centre (sphere) of each TCR variable domain from the three TCR-pMHC structures (color coding as in **a**). **d** Structure of the 6218αC TCR (purple) in complex with PA4C/H2-D$^b$ (peptide in purple, MHC in white) showing a S-S bond formed at the interface. **e** Structure of the 6218 TCR (gold) in complex with PA4C peptide (gold) presented by H2-D$^b$ (white). See Supplementary Fig. 7 for electron density maps and further comparison of PA/H2-D$^b$ and PA4C/H2-D$^b$ structures.

## S-S bonding between TCR and pMHC proteins in solution

To test whether the potential for S-S bond formation affects the dynamics of the 6218αC TCR-PA4C/H2-D$^b$ interaction in solution, we used surface plasmon resonance (SPR). PA4C/H2-D$^b$ had a higher affinity for the 6218 TCR than the 6218αC TCR based on equilibrium constant ($K_{eq}$) values of 12.3 μM and 25.2 μM, respectively (Supplementary Fig. 8a-c). However, after PA4C/H2-D$^b$ injection ended, the 6218 TCR sensorgram returned to baseline, as expected, whereas the 6218αC TCR sensorgram decreased only partially and did not return to baseline (Fig. 6a). This suggested briefly exposing the 6218αC TCR to PA4C/H2-D$^b$ produced distinct short-lived and long-lived subpopulations of 6218αC TCR-PA4C/H2-D$^b$ complexes. Addition of dithiothreitol (DTT), which antagonises S-S bond formation, caused the 6218αC TCR sensorgram to return to baseline after PA4C/H2-D$^b$ injection ended, demonstrating the long-lived subpopulation of 6218αC TCR-PA4C/H2-D$^b$ complexes was dependent on S-S bond formation (Fig. 6b). While long-lived 6218αC TCR-PA4C/H2-D$^b$ complexes persisted for >1 h (Supplementary Fig. 8d), the half-life of short-lived 6218αC TCR-PA4C/H2-D$^b$ complexes was

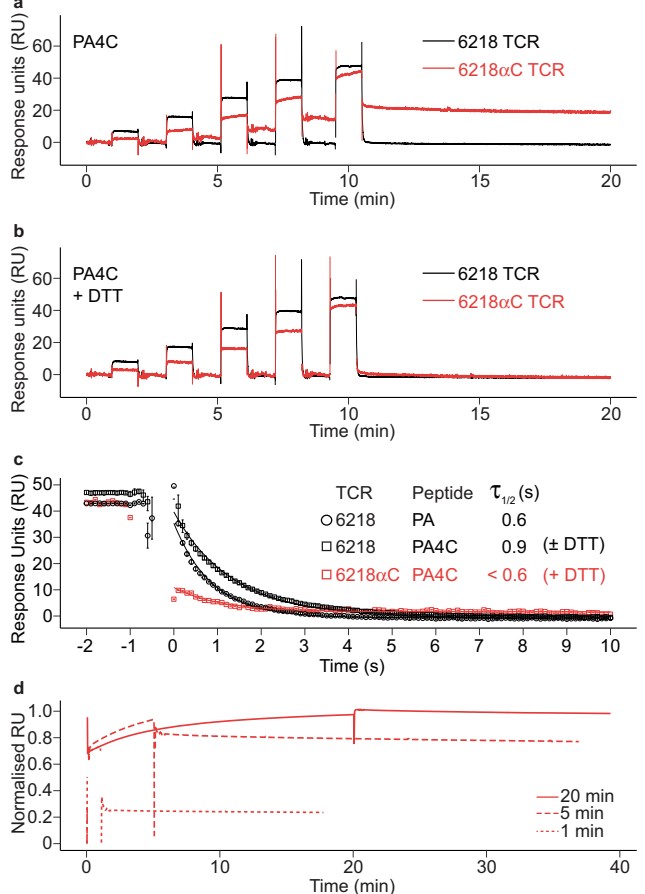

**Fig. 6 | S-S bonding between TCR and pMHC proteins in solution. a** Persistent binding of PA4C/H2-D[b] to 6218αC TCR. SPR sensorgrams of immobilized 6218 TCR (black) or 6218αC TCR (red) exposed to PA4C/H2-D[b] sequentially injected at increasing concentrations. **b** Abrogation of persistent binding by reducing agent (DTT). PA4C/H2-D[b] monomers were incubated in 2 mM DTT overnight before SPR analysis as in **a**. **c** Short half-life of short-lived 6218αC TCR-PA4C/H2-D[b] complexes. SPR data for a 12-s interval spanning the transition from pMHC monomer injection to buffer injection. Some data points between −0.5 s and 0 s are outside the limits of the y-axis. Sensorgrams are aligned so that the onset of measurable dissociation occurs at time = 0.1 s. Error bars show the range of 2 experiments for 6218 TCR-PA/ H2-D[b], 3 experiments for 6218 TCR-PA4C/H2-D[b] (2 without DTT and 1 with DTT) and 1 experiment for 6218αC TCR-PA4C/H2-D[b] (with DTT). **d** Progressive S-S bond formation over a timescale of minutes. SPR sensorgrams for immobilised 6218αC TCR exposed to 100 μM PA4C/H2-D[b] for 1 min (dotted line), 5 min (dashed line) or 20 min (solid line) followed by injection of buffer only. Source data are provided as a Source Data file.

< 1 s (Fig. 6c). Increasing the duration of injection from 1 min to 5 min or 20 min increased the proportion of binding that persisted after PA4C/H2-D[b] injection ended (Fig. 6d). As short-lived 6218αC TCR-PA4C/H2-D[b] complexes turned over within seconds, whereas long-lived complexes accumulated on a timescale of minutes, most associations between the 6218αC TCR and PA4C/H2-D[b] did not result in S-S bond formation. We noted the P4-Cys rotamer in the S-S bond was distinct from the P4-Cys rotamer in the unbound PA4C/ H2-D[b] structure and in the 6218 TCR-PA4C/H2-D[b] complex (Fig. 5b, d, e). It is conceivable that a requirement for the P4-Cys to adopt a specific rotamer limits the frequency of S-S bond formation, at least in solution. Altogether, these data are consistent with a two-phase TCR-pMHC interaction in which noncovalent bonds determine the affinity of the association phase, reflected in the $K_{eq}$ value, and subsequent S-S bond formation marks the transition to a long-lived covalent phase.

## TCR affinity for pMHC determined by noncovalent bonds

To test whether the potential for S-S bond formation affects TCR affinity for pMHC, we tried to engineer lower affinity TCR-pMHC interactions by making substitutions in the peptide. As PA/H2-D[b] and a variant ligand with Ala at P7 bound to mutually exclusive CD8[+] T cell populations[42], we predicted P7 substitutions would result in lower affinity binding to the 6218 and 6218αC TCRs. In cell-based pMHC tetramer binding assays, P7 substitutions in the PA4C peptide resulted in a gradation of binding levels, including an absence of detectable binding; however, at each step in the gradation we obtained similar results for the 6218 and 6218αC TCRs (Fig. 7a). SPR also revealed each P7 substitution decreased binding to both TCRs (Fig. 7b and Supplementary Fig. 8c). Modified assay conditions did reveal evidence of S-S bond formation, however, as demonstrated by reduced dissociation of P4-Cys-containing pMHC tetramers from cells expressing the 6218αC TCR (Fig. 7c and Supplementary Fig. 9) and SPR revealed persistent binding of P4-Cys-containing pMHC monomers to the 6218αC TCR after long-duration injections (Fig. 7d). The potential for S-S bond formation thus had an unremarkable effect on the initial affinity of TCR association with pMHC.

## T cell activation by low-affinity TCR-pMHC interactions

To investigate effects of S-S bond formation on T cell activation, we transduced a T cell line, 5KC, which does not express an endogenous TCRβ chain[43], with mouse CD8αβ and either the 6218 or 6218αC TCR. Transduced cells were cocultured with DC2.4 cells[44], which express H2-D[b] naturally, and peptide-induced T cell activation was evaluated by measuring IL-2 in the supernatant. The $EC_{50}$ of the PA4C peptide when used to stimulate cells expressing the 6218αC TCR was 1 nM, whereas it took 49 nM of PA4C peptide to induce a similar response in the 6218 TCR cells (Fig. 8a). Furthermore, the Hillslope (h) value, which reflects the steepness of the dose-response curve, was greater for cells expressing the 6218αC TCR (Fig. 8a). These results demonstrate S-S bond formation confers a marked increase in T cell antigen sensitivity that is not due to an increase in the initial TCR affinity for pMHC.

Cells expressing the 6218 TCR were not activated by any peptide with a P7 substitution, whereas cells expressing the 6218αC TCR were activated by all peptides with a P7 substitution if P4-Cys was present (Fig. 8a). These results extend the evidence that covalent antigen recognition is characterized by increased antigen sensitivity compared to noncovalent antigen recognition. They also reveal that T cell antigen specificity is inferior in covalent antigen recognition compared to noncovalent antigen recognition. However, when used to stimulate cells expressing the 6218αC TCR, P4-Cys-containing peptides did result in a gradation of $EC_{50}$ and h values, which corresponded to the binding hierarchy observed in tetramer and SPR assays (Fig. 8b). This suggests the initial TCR-pMHC affinity does contribute to antigen specificity even during covalent antigen recognition in T cells.

## Discussion

Interactions between TCRs and pMHC ligands are extremely diverse, yet physiological TCR-pMHC interactions that involve covalent bonding were unknown. Here we demonstrate the potential for TCR-peptide S-S bond formation has little effect on the initial TCR affinity for pMHC; however, after S-S bond formation, the long lifetime of covalently bound TCR-pMHC complexes mimics a high-affinity interaction. Covalent TCR-pMHC interactions enable T cells to react to pMHC ligands with subthreshold affinities for the TCR. This mechanism occurs naturally in the thymic cortex and results in immature T cells undergoing deletion or differentiation into CD8αα IEL. These findings elucidate a novel mechanism of physiological T cell activation that has applications in basic immunology and potentially in immunotherapy.

As the kinetic proofreading model posits that T cell activation requires TCR-pMHC complexes to persist long enough to activate TCR

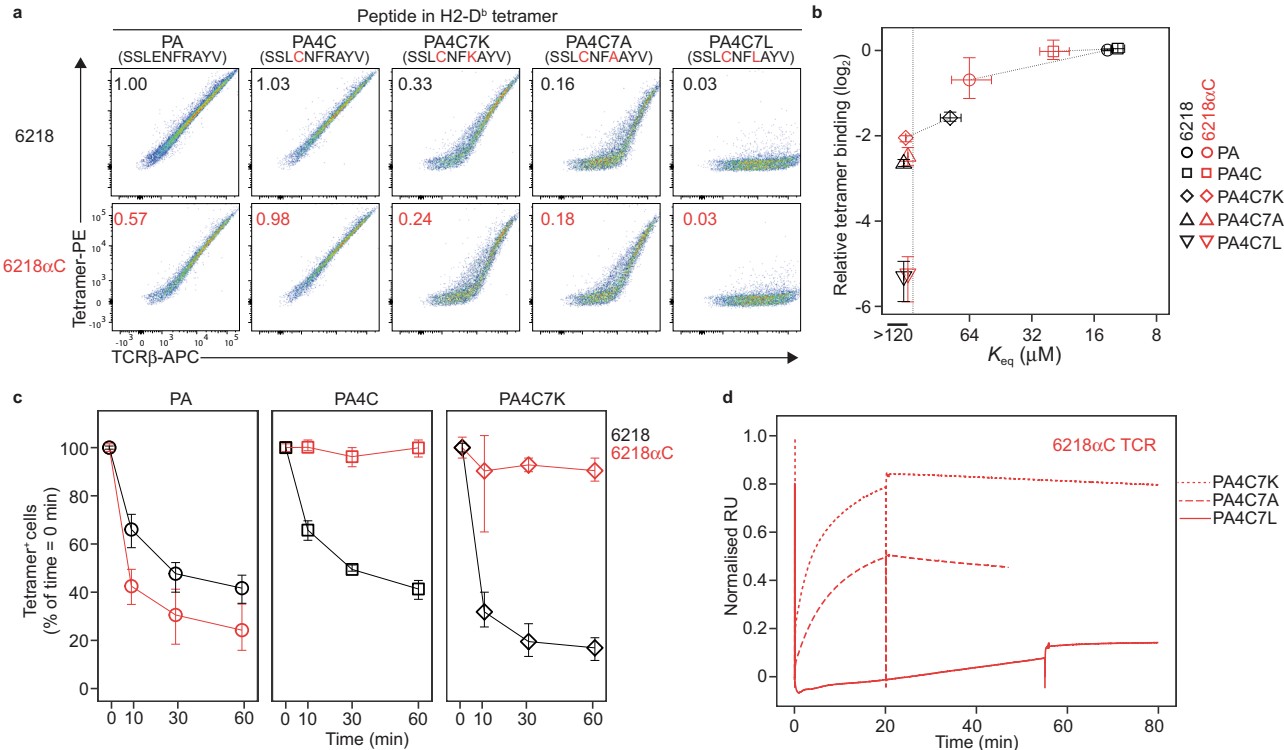

**Fig. 7 | S-S bonding across a wide range of TCR affinities for pMHC.**
**a** Substitutions at P7 of the PA4C peptide decrease TCR binding by pMHC tetramers. Plots show FACS phenotypes of TCR transfectants expressing the 6218 or 6218αC TCR stained with anti-TCRβ and tetramers of H2-Dᵇ presenting the indicated peptide (top). Numbers on plots show the tetramer mean fluorescence intensity (MFI) divided by TCRβ MFI, normalized to the 6218 TCR-PA/H2-Dᵇ sample.
**b** For the indicated TCR-peptide/H2-Dᵇ combinations (right), symbols show the mean tetramer binding, calculated as in **a**, as a function of the $K_{eq}$ determined by SPR (see Supplementary Fig. 8c), with dotted lines connecting measurements from the same pMHC. Error bars show the range of 3-6 samples from 2-3 experiments (y-axis) or the standard error of the mean of 2 experiments with a total of 4 samples per combination (x-axis). **c** S-S bond formation inhibits tetramer dissociation. TCR

transfectants expressing 6218 or 6218αC and stained with pMHC tetramers as in **a** were washed and resuspended in buffer containing 25 μg/mL anti-H2-Dᵇ/Kᵇ to prevent tetramer rebinding for 10 min, 30 min or 60 min before FACS analysis. Graphs show the tetramer⁺ cell frequency as a percentage of the corresponding sample without anti-H2-Dᵇ/Kᵇ (time = 0 min). Symbols show the mean, and error bars the range, of 4 samples per condition compiled from 2 experiments.
**d** Persistent interactions between P4-Cys-containing pMHC monomers and the 6218αC TCR. SPR sensorgrams show binding to the 6218αC TCR during and after 20-min injections of PA4C7K/H2-Dᵇ (dotted line) or PA4C7A/H2-Dᵇ (dashed line), or a 50-min injection of PA4C7L/H2-Dᵇ (solid line), all at 100 μM. Source data are provided as a Source Data file.

signaling[45], this model is sufficient to explain why long-lived, covalently bound TCR-pMHC complexes elicit T cell activation. To explain suboptimal T cell responses to low amounts of pMHC ligand by certain affinity-enhanced TCRs[46–48], it was postulated that a TCR-pMHC complex can only signal for a limited amount of time, and antigen sensitivity is optimised by intermediate-affinity TCRs that unbind and rebind pMHC repetitively[47]. Our results demonstrate rebinding is not necessary for T cell activation. Suboptimal T cell activation in response to affinity-enhanced TCR-pMHC interactions may thus be due to the fast on-rates, and not the long lifetimes, of those interactions[46–48]. An intermediate on-rate may optimise T cell sensitivity to low doses of pMHC ligands by limiting the rate at which TCR-pMHC complexes are removed from the immunological synapse via internalisation into the T cell[49] or the antigen-presenting cell[50]. Covalent antigen recognition provides a novel framework to dissect T cell activation in a setting where TCR-pMHC complex lifetime and rebinding are not variables and T cells can be activated by lower amounts of pMHC and by pMHC ligands with lower affinity for the TCR.

Strong TCR signaling induces deletion of thymocytes before or after they upregulate CCR7[51] and CCR7⁻ thymocytes that survive strong TCR signaling differentiate into Type A IELp[27,52]. Our findings clarify the mechanism underlying the enrichment of Cys-containing CDR3 in the Type A IELp TCR repertoire[30,32]. Interactions between Cys-containing CDR3 and Cys-containing pMHC self-antigens lead to S-S

bond formation. Covalently bound TCR-pMHC complexes resist rupture and elicit strong TCR signaling in CCR7⁻ thymocytes, which respond by undergoing deletion or Type A IELp differentiation. Despite expressing 6218αC, 6218βC, or other TCRs with a Cys-containing CDR3, *Zap70^{mrd/mrt}* thymocytes matured into T-conv cells, demonstrating the physiological outcomes of long-lived TCR-pMHC interactions require an intact TCR signaling pathway. The reproducibility of T cell fates associated with Cys-containing CDR3 is likely due to the extreme sensitivity of pre-selection thymocytes to pMHC ligands[53–55] and the ability of S-S bonds to form even when the initial TCR affinity for pMHC is low. Together with the finding that most Type A IELp reside in the thymic cortex[27], our evidence that a Cys-containing CDR3 triggers deletion in the thymic cortex indicates thymocytes diverge towards these fates in the thymic cortex.

As therapies for autoimmunity, T-reg cells can be purified, expanded, genetically modified and reinfused safely, but efficacy has been modest[56]. Organ-specific autoimmunity arises in individuals who lack a pMHC self-antigen that can induce a protective organ-specific T-reg population[57], suggesting the lack of an effective pMHC ligand is a problem in T-reg therapy. Our findings suggest diverse Cys-containing pMHC self-antigens are presented to T cells in vivo and may be effective pMHC ligands for T-reg therapy. Due to the anti-inflammatory function of T-reg, some TCR cross-reactivity may be acceptable and even beneficial[58]. Thus, engineered TCRs that form a S-S bond with

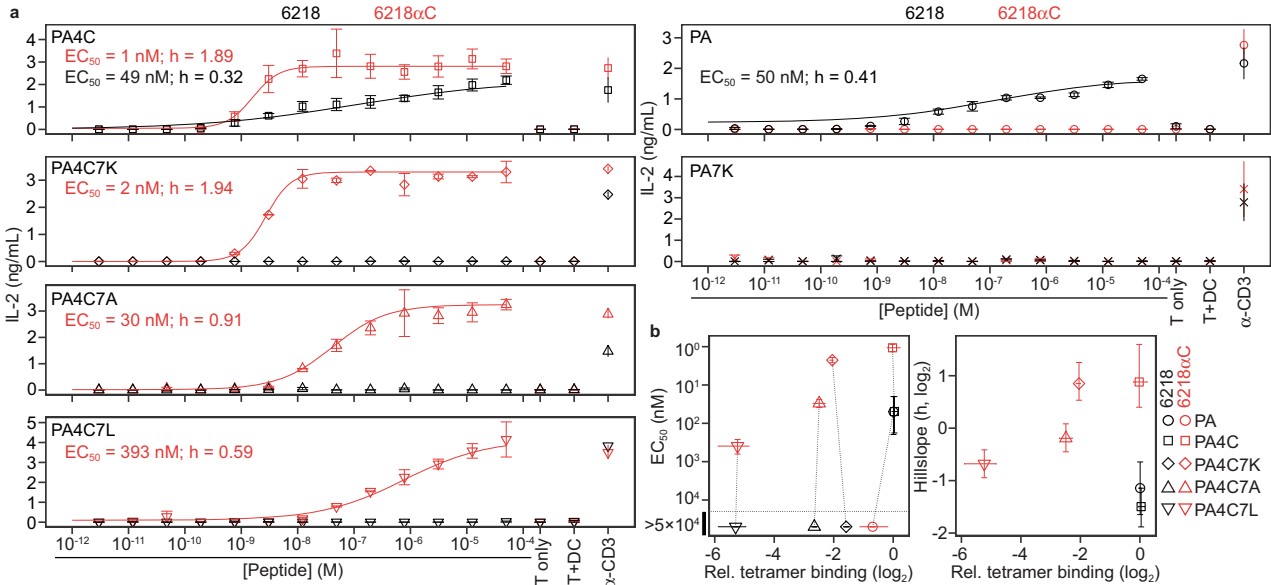

**Fig. 8 | T cell activation due to TCR-peptide S-S bonding. a** Supernatant IL-2 concentration after coculture of 5KC T cells expressing 6218 or 6218αC and DC2.4 cells with graded concentrations of the peptide denoted in the upper left of each graph. Control cultures included 5KC cells with 50 μM peptide (T only), 5KC and DC2.4 cells without peptide (T + DC), and 5KC cells with plate-bound anti-CD3 (α-CD3). Symbols show the mean, and error bars the range, of 2 wells per condition from one experiment, representative of at least 2 experiments. HillSlope (h) values were determined using nonlinear regression. For curves that did not reach a plateau, the reported $EC_{50}$ values provide a minimum estimate of the $EC_{50}$. **b** For each TCR/peptide combination, graphs show relative tetramer binding (x-axes, plotted as in Fig. 7b) against $EC_{50}$ (left) or h (right) values, which were determined from a total of at least 4 dilution series per peptide analyzed in at least 2 experiments. Error bars indicate the 95% confidence intervals of $EC_{50}$ and h; dotted lines connect measurements from the same pMHC. Source data are provided as a Source Data file.

Cys-containing pMHC antigens may confer potent T-reg activation and superior immunosuppression in future therapies for autoimmune disease.

# Methods

## Mice
B6.*Rag1^−/−* *mice* (*Rag1^tm1Bal^* [59]) were purchased from Bioservices Kew, Walter and Eliza Hall Institute of Medical Research (Melbourne, Australia). B6 mice, and mice carrying *Tcra^tm1Mom^* [60], *B2m^tm1Jae^* [40], *H2-Aa^tm1Blt^* [39], *Zap70^mrd^* [38], *Zap70^mrt^* [38], BCL2-tg (*Tg^(Vav-BCL2)1Jad^*)[61], Yae62β-tg [41], *Foxp3^tm2Ayr^* (Foxp3^GFP^)[62], all on the B6 genetic background, were bred (intercrossed in some cases) and housed in specific-pathogen-free environments at 18-24 °C and 40-70 % humidity with a lighting cycle of 7 a.m. to 7 p.m. light (below 350 lux) and 7 p.m. to 7 a.m. darkness, at the Australian Phenomics Facility, Canberra, or at Monash University, Melbourne. Genotyping was performed on genomic DNA extracted from ear or tail biopsies by the Australian Phenomics Facility (Canberra, Australia) or Transnetyx Inc. (Cordova TN). Except for Fig. 3a and Supplementary Figs. 2, 3, which show data from mice that were co-housed, mice in different experimental groups were housed separately. Mice were euthanised by carbon dioxide inhalation. All procedures were performed in accordance with protocols approved by The Animal Experimentation Ethics Committee of the Australian National University (A2014/62 and A2018/06) or the Monash University Animal Experimentation Ethics Committee (MARP/2015/64).

## TCR-retrogenic mice
DNA encoding the 6218 TCRα and TCRβ genes separated by the "cleavable" P2A peptide were synthesised (Genscript Biotech, Piscataway, NJ). Cys-encoding codons were introduced into the CDR3α (Ser110α replaced by Cys110α) or CDR3β (Gly109β replaced by Cys109β) sequence using PCR mutagenesis. DNA constructs were cloned into the pMSCV-IRES-GFP II (pMIG II) vector, which encodes GFP under the control of an internal ribosomal entry site [35]. *Rag1^−/−* BM

cells were retrovirally transduced in vitro, as described previously [35,63]. B6 BM was depleted of T cells using a mouse CD3ε MicroBead Kit (Miltenyi, Bergisch Gladbach, Germany, Cat. no. 130-094-973) and an autoMACS machine (Miltenyi). *Rag1^−/−* BM cells and T cell-depleted B6 BM cells were mixed 1:1 before i.v. injection of > 2 × 10^6 cells per *Rag1^−/−* or B6 recipient, which had been irradiated with x-rays (two doses of 5 Gy given 4 h apart) earlier in the day. *Tcra^−/−* BM donors and recipients, or *Zap70^mrd/mrt^* BM donors and recipients, were used to make TCR-retrogenic mice with the same protocol, except that T cell-depleted B6 BM cells were not administered in these cases.

## TCR and peptide/MHC synthesis
DNA fragments, optimised for bacterial expression, encoding the mouse variable domains of the 6218 TCR [33] and human constant domains were cloned separately into a pET30 expression vector (Genscript Biotech). In addition, the 6218αC TCR was cloned similarly with Ser110α replaced by Cys110α (see Fig. 1a). The 6218 TCRα, 6218 TCRβ, and 6218αC TCRα chains were expressed in BL21 *E.coli* cells (American Type Culture Collection, Manassas, VA, Cat. no. PTA-5976) separately as inclusion bodies. Functional and soluble TCRs were produced by refolding equal amounts of α- and β-chains into a refold mixture as described [33] for 3 days, followed by dialysis into 10 mM Tris-HCl pH 8.0. The refolded TCRs were purified by anion exchange and size-exclusion chromatography. Human β2m and H2-D^b heavy chain (residues 1–274) fused to a BirA-substrate peptide (AviTag, Avidity, LLC, Aurora, CO) were expressed separately in BL21 *E. coli* cells and extracted as inclusion bodies. These inclusion bodies (30 mg H2-D^b and 10 mg of β2m) were refolded with 4 mg of peptide (Genscript Biotech) for 3 h. Folded pMHC complexes were purified by anion exchange and size exclusion chromatography, biotinylated using BirA biotin ligase with the addition of D-biotin (Astral Scientific, Sydney, Australia, Cat no. BIOBB0078), and tetramerised by addition of PE-Streptavidin (BioLegend, Cat. no. 405204) at a 4:1 molar ratio. DNA encoding BirA biotin ligase was cloned into a pcDNA3.1 expression vector with a His-tag (Genscript Biotech), and the protein was

expressed in BL21 *E. coli* cells, then purified using Ni-NTA agarose beads (Machery-Nagel, Düren, Germany, Cat. no. 745400.100).

## TCR transfectants

293 T (human embryonic kidney cell line) cells maintained in Dulbecco's modified eagle medium supplemented with 10% fetal calf serum (Gibco, Amarillo, TX, Cat. no. 10437028), 2 mM L-glutamine (Gibco, Cat. no. 25030149), 1 mM sodium pyruvate (Gibco, Cat. no. 11360070), 100 µM Non-essential amino acids (Gibco, Cat. no. 11140050), 5 mM HEPES buffer (Gibco, Cat. no. 15630080), 55 µM 2-mercaptoethanol (Gibco, Cat. no. 21985023), 100 units/mL penicillin and 100 µg/mL streptomycin (Gibco, Cat. no. 15140122) (cDMEM) were mixed with FuGENE 6 Transfection Reagent (Promega Corporation, Cat. no. E2691) and two distinct pMIG II expression vectors encoding GFP downstream of an internal ribosome entry site[35]. One plasmid (pMIGII/CD3) contained DNA sequences encoding mouse CD3γ, CD3δ, CD3ε and CD3ζ separated by the 2 A peptide[35]. Another plasmid contained DNA sequences encoding the TCRα and TCRβ chains of the 6218, 6218αC or 6218βC TCR separated by DNA encoding the 2 A peptide. 48 h after transfection, TCR transfectants were incubated with PE-conjugated pMHC tetramers for 1 h at RT. Cells were washed, incubated with APC anti-TCRβ (BioLegend, Cat. no. 109212) and LIVE/DEAD™ Fixable Aqua Dead Cell Stain (Thermofisher, Waltham, MA, Cat. no. L34957) for 30 min, then washed before flow cytometry analysis. For tetramer dissociation assays, TCR transfectants were stained with 5 µg/mL pMHC tetramers for 1 h at RT, then washed and incubated for 10, 30, or 60 min with 25 µg/mL anti-H2-D$^b$/K$^b$ (BD Biosciences, clone 28-8-6, Cat. no. 553575) to prevent tetramer rebinding, then washed and stained with anti-TCRβ-APC and LIVE/DEAD™ Fixable Aqua Dead Cell Stain before flow cytometric analysis.

## Surface plasmon resonance

Each TCR was immobilized onto a CM5 sensor chip via amine coupling. 6218αC TCRs formed disulfide-bonded homodimers, which did not bind to PA/H2-D$^b$ or PA4C/H2-D$^b$ unless TCR dimerisation was removed using a 10-min injection of 1 mM DTT at 10 µL/min. The DTT-treated flow cell was equilibrated for 3 h in running buffer to reach a steady baseline. pMHC was flowed over the TCRs in a series of 1-min injections with increasing pMHC concentrations (1.5 µM, 4.4 µM, 13.3 µM, 40 µM and 120 µM, or, in the cases of PA4C7L/H2-D$^b$ and PA7L/H2-D$^b$, 3 µM, 8.8 µM, 26.6 µM, 80µM and 240 µM) using a 1 in 3 dilution. Data were exported using BIAevaluation 3.0 Software and $K_{eq}$ values were calculated by fitting steady-state data points into a 1:1 Langmuir binding model in Prism version 9 (GraphPad Software, La Jolla, CA). Error estimates associated with the $K_{eq}$ values represent the standard error of the mean from two independent experiments performed in duplicate. To induce dissociation of PA4C/H2-D$^b$ from the 6218αC TCR, samples of PA4C/H2-D$^b$ were treated with 2 mM DTT overnight at 4 °C before dilution into running buffer without the presence of DTT. In assays involving pMHC injection times > 1 min, sequential 1-min injections of a negative control pMHC (Influenza Virus NP$_{265-274}$/HLA-A*03) and a positive control pMHC (PA/H2-D$^b$) preceded the test pMHC monomer injections of 5, 20, or 50 min; in these assays, the concentration of all pMHC monomers was 100 µM. For assays involving pMHC injection times > 1 min, to account for differences in the amount of immobilized 6218αC TCR between sensor chips, we normalized the Response Units (RU) using the formula, $K_{eq} = [6218αC$ TCR] × [PA/H2-D$^b$] / [6218αC TCR-PA/H2-D$^b$]. As $K_{eq} = 64.1$µM and [PA/H2-D$^b$] = 100 µM, the "Normalized RU" value = 1 (corresponding to an estimated 6218αC TCR occupancy of 100%) was set as 1.641 × the RU detected at equilibrium when the PA/H2-D$^b$ positive control was flowed over the 6218αC TCR. Experiments were conducted at 25 °C on a BIAcore T100 (GE Healthcare, Chicago, IL) with 10 mM Tris-HCl, pH 8, 150 mM NaCl, 0.005% surfactant P20 containing 0.1% bovine serum albumin (Merck & Co., Kenilworth, NJ, Cat. no. 810667) to avoid non-specific binding.

## X-ray crystallography

Crystal screens were set up via sitting-drop, vapour diffusion at 20 °C with a protein:reservoir drop ratio of 1:1, at a concentration of 3 mg/mL in 10 mM Tris-HCl pH 8, 150 mM NaCl. Crystals of 6218-PA/H2-D$^b$, 6218-PA4C/H2-D$^b$ and 6218αC-PA4C/H2-D$^b$ complexes were grown in 20% (w/v) PEG3350, 0.2 M NaF, 0.05 M NaFormate and 3% (w/v) 1,5-Diaminopentane dihydrochloride. Crystals of H2-D$^b$ in complex with PA4C were grown at 3 mg/mL in 20% (w/v) PEG2000, 0.1 M KSCN and 2% (w/v) 2-Methyl-2,4-pentanediol. All crystals were soaked in a cryoprotectant solution containing mother liquor solution with the PEG3350 concentration increased to 30% (w/v) and then flash-frozen in liquid nitrogen. The data were collected using AS GUI on the MX2 beamline at the Australian Synchrotron, part of ANSTO, Australia[64]. The data were processed using XDS (BUILT = 20161205)[65]. Data were scaled and reduced using the Pointless and Aimless program[66] from the CCP4 suite (version 7.0.077). Structures were determined by molecular replacement using the PHASER program (version 2.8.3)[67] from the CCP4 suite (1994) (version 7.0.077) with a model of H2-D$^b$ without the peptide derived from PDB ID: 3PQY[33]. Manual model building was conducted using COOT (version 0.8.9.2)[68] followed by refinement with BUSTER (version 2.10.3)[69]. The final model has been validated using the wwwPDB OneDep System and the final refinement statistics are summarised in Supplementary Table 1. All molecular graphics representations were created using PyMOL (version 2.5.0a0).

## Coculture assay

5KC-73.8.20 (5KC) cells[43] were sorted for loss of CD4 to establish a CD4$^-$ CD8$^-$ cell line and maintained in cDMEM. 5KC cells were transduced to express CD8αβ and either the 6218 or 6218αC TCR encoded in the pMIGII retroviral vector and sorted for equivalent expression of CDαβ and TCRβ. 5 × 10$^4$ transduced 5KC cells were incubated with 10$^5$ DC2.4 mouse dendritic cells[44], in the absence or presence of graded concentrations of peptide or anti-CD3 (10 µg/mL pre-bound to plate overnight), for 16 h. Supernatants were collected and assayed for IL-2 concentration using the BD OptEIA Mouse IL-2 enzyme-linked immunosorbent assay (ELISA) kit. EC$_{50}$ and h values, and their associated 95% confidence intervals, were determined using nonlinear regression (4-parameter dose-response curves) in GraphPad Prism.

## Histology

Thymus samples were immersed in 50% ethanol/5% acetic acid/45% water for 10 min before transfer to 10% neutral buffered formalin overnight. Fixed samples were then exposed to a 4-h cycle of graded ethanols and xylene using a Peloris II Tissue Processor (Leica, Wetzlar, Germany) before embedding in Paraplast wax (P3558, Sigma-Aldrich, St. Louis, MO) and sectioning onto superfrost plus slides at a thickness of 4 µm using a RM2235 microtome (Leica). Primary antibodies, Chicken anti-GFP (ab13970, Abcam, Cambridge, UK) and Rabbit anti-Cytokeratin 14 (K14) (ab197893, Abcam), were used at a dilution of 1 in 200; secondary antibodies, Goat Anti-Chicken IgY A647 (ab150175, Abcam) and Donkey Anti-Rabbit IgG A488 (711-545-152, Jackson ImmunoResearch, West Grove, PA), at 1 in 500. Staining was performed using an Autostainer Link 48 (Dako, Glostrup, Denmark) with the following incubations, each interspersed by one or two 5-min incubations in Wash Buffer (K8007, Agilent, Santa Clara, CA): Target Retrieval Solution S1699 (Agilent) at 98 °C for 30 min, Protein Block X0909 (Agilent) at room temperature (RT) for 60 min, primary antibodies at RT for 60 min, secondary antibodies at RT for 60 min, DAPI at RT for 15 min. Imaging was performed on an Olympus VS120 virtual slide microscope with a UPLS APO 20× lens with 0.75 NA and captured with an Olympus XM10 digital camera. A DAPI/FITC/CY5 filter set was

used with the same exposure settings for all sections. Fiji[70] was used to quantify the areas of cortex and medulla, defined as K14⁻ and K14⁺, respectively, and to count GFP⁺ cells. As the anti-GFP antibody stained some capsular and subcapsular areas in negative control B6 thymic sections, the outer ~ 50 μm regions of thymic sections were excluded from quantification. In thymic sections from 4 TCR-retrogenic mice (3 in the 6218 group and 1 in the 6218αC group) the number of GFP⁺ cells per mm² was within 2 standard deviations of the mean of thymic sections from negative control B6 mice ($n = 6$); these samples were excluded. The software used to make images for Fig. 2 was cellSens Dimension version 4.1 (Olympus Corporation, Tokyo, Japan).

## T cell sorting

Whole thymus or spleen suspensions were prepared by pushing organs through a 70 μm sieve in sort buffer (PBS containing 2% v/v heat-inactivated fetal calf serum and 2 mM EDTA). Small intestine was first cut longitudinally and then into pieces ~0.5 cm long while being kept moist with washing medium (WM, DMEM containing 2.5% v/v heat-inactivated bovine serum and 10 mM HEPES) and placed in a 50 mL tube containing ~15 mL ice-cold WM. Intestinal contents were removed by cycles of vortexing for 5 s, then removing the supernatant by using a strainer to retain intestinal tissue and resuspending in 15 mL WM, until supernatant was clear. Tissue pieces were then incubated for 15 minutes at 37 °C with gentle rotation in dissociation buffer (calcium- and magnesium-free PBS containing 5% v/v heat-inactivated bovine serum plus 2 mM EDTA). After vortexing for 15 s, tissue pieces were removed using a strainer and discarded, while the supernatant was passed through a 70 μm sieve, pelleted by centrifugation, resuspended in 5 mL of 40% Percoll and overlaid onto 5 mL 80% Percoll in a 15 mL tube. After centrifugation for 20 min at 900 g at 20 °C, material at the interface was collected, transferred to a fresh 15 mL tube containing 10 mL of sort buffer, pelleted and incubated with antibodies as described below (see Supplementary table 2 for details of antibodies and Supplementary Figs. 4, 5 for all T cell sorting gating strategies). For CCR7 staining of thymocytes, suspensions were incubated for 60 min at 37 °C in 1 mL pre-warmed sort buffer containing phycoerythrin (PE)-conjugated anti-CCR7 (BioLegend, San Diego, CA, Cat. no. 120105). For staining of other cell surface markers, each thymus or spleen sample was incubated in 1 mL sort buffer, and each small intestinal sample was incubated in 0.5 mL sort buffer, containing fluorescently conjugated antibodies for 30 min at 4 °C. After washing, cells were passed through a 40 μm sieve before using an Influx Cell Sorter instrument (Becton Dickinson, Franklin Lakes, New Jersey) to sort T cell subsets (typically $5 \times 10^4$ cells per sample) into 1.5 mL Eppendorf tubes containing 350 μL Buffer RLT from the RNeasy Mini Kit (Qiagen, Hilden, Germany, Cat. no. 74106). Samples were then frozen by pushing into dry ice and stored at −80 °C until RNA isolation.

## TCR sequencing

RNA was isolated using the RNeasy Mini kit with an elution volume of 22 μL, from which 12 μL were used to synthesise cDNA with the QuantiTect Reverse Transcription Kit (Qiagen, Cat. no. 205311). Using 5 μL of cDNA per reaction, TCRβ transcripts were PCR amplified using a Q5® High-Fidelity PCR Kit (New England BioLabs, Ipswich, MA, Cat. no. E0555L) and a mix of 19 *Trbv*-specific forward primers and a single *Trbc*-specific reverse primer and TCRα transcripts were PCR amplified using a mix of 23 or 24 *Trav*-specific forward primers and a single *Trac*-specific reverse primer (Supplementary Table 3, all purchased from GeneWorks, Adelaide, Australia). Forward and reverse primers had distinct 5' overhang adapter sequences that enabled addition of sample-specific indices and P5/P7 sequencing adapters in a second PCR using the Nextera XT DNA Library Preparation Kit (Illumina, San Diego, CA, Cat. no. FC-131-1096). Before the second PCR, AMPure XP magnetic beads (Beckman Coulter, Brea, CA, Cat. no. A63881) were used to enrich amplicons > 100 bp. Conditions for the first PCR were

98 °C for 5 min, 20 cycles of 98 °C for 10 s, 60 °C for 30 s and 72 °C for 30 s, followed by 72 °C for 2 min and for the second PCR were 72 °C for 3 min and 98 °C for 30 s, 20 cycles of 98 °C for 10 s, 63 °C for 30 s and 72 °C for 30 s, followed by 72 °C for 1 min. After determining amplicon concentrations using a QIAxcel capillary electrophoresis machine (Qiagen), equimolar amounts of amplicons from up to 270 samples were pooled into a single tube, concentrated using AMPure XP magnetic beads and then 300-500 bp amplicons were gel-purified before sequencing on a NextSeq machine (Illumina), with a short read 1 of 6 bases followed by a read 2 of 145 bases.

## TCR sequence analysis

Sequences were aligned to mouse TCR genes using molecular identifier groups-based error correction (MIGEC) software (version 1.2.6)[71]. Subsequent analyses were performed using RStudio software (version 2022.02.3 Build 492). Sequences with a CDR3 that was out-of-frame or contained a stop codon were excluded. A clone was defined as a unique combination of *Trav* or *Trbv* gene and CDR3 nucleotide sequence. Each clone was counted only once per sample regardless of its number of reads. CDR3 length was determined using the CDR3-IMGT definition, which excludes the conserved N-terminal Cys and C-terminal Trp or Phe from the CDR3[72]. For a CDR3 sequence of $n$ amino acids, the amino acid at the largest position not greater than ($n/2 + 1$) was defined as the middle CDR3 position (apex). The cysteine index for each sample equals the percentage of clones with Cys within 2 positions of the CDR3 apex. Sequences detected only once or twice in any given sample were excluded from cysteine index calculations. For the sample that had 0 clonotypes with Cys within two positions of the CDR3 apex, the cysteine index was defined as the reciprocal of the number of clones in the sample, expressed as a percentage. As *Trbv1* sequences can have a germline-encoded Cys at CDR3 position 2, which is within 2 positions of the apex of CDR3 sequences < 8 amino acids long, we excluded *Trbv1* sequences with a CDR3 length < 8 amino acids. For analyses of Yae62β-tg mice, we used a published dataset of TCRα sequences expressed in Yae62β-tg mice[30] to compile 4 reference TCR catalogs [(i) Type A IELp and CD8αα IEL; (ii) CD4⁺ T-conv; (iii) CD8⁺ T-conv and (iv) Foxp3⁺ T-reg] by aggregating all clones detected in thymic or peripheral samples of a given T cell lineage. We then defined a TCR clonotype as a unique combination of *Trav* gene and CDR3 amino acid sequence. Each clonotype's intra-lineage frequency equals the number of clones that encode the clonotype divided by the total number of clones in the TCR catalog. Each clonotype's "Distribution in reference TCR catalogs" equals its intra-lineage frequency or frequencies divided by the sum of its intra-lineage frequencies across the 4 TCR catalogs, expressed as a percentage.

## Flow cytometry

For quantification of TCR-signalled nascent thymocytes, mice were injected i.v. with 0.25 mg 5-ethynyl-2'-deoxyuridine (EdU) in 0.2 mL PBS 3 d before analysis and thymocytes were stained for CCR7 as for T cell sorting, except that biotin anti-mouse CD197 (CCR7) was used. For analysis of TCR-retrogenic mice, single-cell suspensions of thymus, spleen or small intestine samples were prepared, and CCR7 staining was performed on thymocytes, as for T cell sorting. Otherwise, or additionally, up to $4 \times 10^6$ cells were incubated in 50 μL FACS buffer (PBS containing 2% v/v heat-inactivated bovine serum and 0.01% m/v sodium azide) for 30 min at 4 °C containing assortments of the anti-mouse antibodies: Brilliant Violet 510 anti-TCRβ (BioLegend, Cat. no. 109233), Alexa Fluor 700 anti-CD4 (BioLegend, Cat. no. 100430), APC/Fire anti-CD8α (BioLegend, Cat. no. 100766), PE/Cy7 anti-CD8β.2 (BioLegend, Cat. no. 140416), Brilliant Violet 421 anti-CD279 (PD-1) (BioLegend, Cat. no. 135218), PE anti-NK1.1 (Miltenyi, Cat. no. 130-102-400), VioBlue anti-CD45 (Miltenyi, Cat. no. 130-102-430) and propidium iodide solution (BioLegend, Cat. no. 421301). Samples were then washed in FACS buffer and data were acquired, or the cells were fixed

and permeabilised using the Foxp3 / Transcription Factor Staining Buffer Set (Thermo Fisher Scientific, Waltham, MA, Cat. no.00-5523-00), then incubated with anti-Helios-Pacific Blue (BioLegend, Cat. no. 137220). Samples were then processed using the Click-iT® EdU Flow Cytometry Assay Kit (Thermo Fisher Scientific, Cat. no. C10420) following the manufacturer's instructions except that Click-iT® EdU buffer additive (Component G) was used at one-fifth of the concentration recommended. Samples were then washed in FACS buffer and incubated with PE-Vio770-conjugated anti-CD24 (Miltenyi, Cat. no. 130-102-736) and PE-streptavidin (BioLegend, Cat. no. 405204). After washing in FACS buffer, data were acquired with Fortessa or LSR II flow cytometers (Becton Dickinson) and analyzed using FlowJo software (FlowJo LLC, Ashland, Oregon).

### Data visualization and statistical analyses
The 'tidyverse' and 'stringr' packages were used to perform TCR sequence analyses and produce graphs in RStudio. Statistical analyses were performed using GraphPad Prism. Figures were made using Adobe Illustrator (version 25.4.5, Adobe Systems Inc.) or Microsoft PowerPoint. Supplementary Fig. 3 was partly generated using Servier Medical Art, provided by Servier, licensed under a Creative Commons Attribution 3.0 unported license.

### Reporting summary
Further information on research design is available in the Nature Research Reporting Summary linked to this article.

## Data availability
The TCR sequencing data generated in this study have been deposited in the NCBI Short Read Archive under BioProject ID "PRJNA733833" for Fig. 3 and "PRJNA734126" for Supplementary Fig. 6. The crystal structure coordinates have been deposited in the Protein Data Bank (PDB) under the following accession codes: "7N5Q" for PA4C/H2-D$^b$, "7N4K" for 6218-PA/H2-D$^b$, "7N5P" for 6218-PA4C/H2-D$^b$, and "7N5C" for 6218αC-PA4C/H2-D$^b$. All data are included in the Supplementary Information or available from the authors upon reasonable requests, as are unique reagents used in this Article. The raw numbers for charts and graphs are available in the Source Data file whenever possible. Source data are provided with this paper.

## Code availability
Code, processed TCR sequence data, and examples of expected output are available at https://doi.org/10.5281/zenodo.6842270[73].

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

## Acknowledgements

We thank the Monash University platforms for Animal Research, Histology, FlowCore, Macro Molecular Crystallisation facility, and Micromon for their services. We thank Dr Philippa Marrack (National Jewish Health, Denver, CO) for the 5KC cell line; Dr Ken Rock (University of Massachusetts, Worchester, MA) for the DC2.4 cell line; Dr Nicholas Gascoigne (National University of Singapore, Singapore) for the mouse CD8αβ constructs; Dr Dario Vignali (University of Pittsburgh, Pittsburgh, PA) for the pMIG II, pMIG II-mouseCD3, pEQ-Pam3(-E), and pVSVG vectors; and the MX team for assistance at the Australian Synchrotron. This research was undertaken in part using the MX2 beamline at the Australian Synchrotron, part of ANSTO, and made use of the ACRF detector. This research was funded by Monash Biomedicine Discovery Institute (SRD), National Health and Medical Research Council [grants 1107464 and 1188589 (SRD), grant 1159272 (SG) and grants 1071916 and 1182086 (NLLG)], and Australian Research Council [grants DP170103631, DP200102776 and FT170100174 (NLLG)].

## Author contributions

TCR and pMHC synthesis: C.S. and A.T.N.; TCR transfectants, P.Z. and J.B.Z.; Surface Plasmon Resonance: C.S.; X-ray crystallography, C.S., A.R-T. and S.G.; Coculture assay, P.Z., J.B.Z. and S.R.D.; TCR-retrogenic mice, P.Z., R.C.W. and S.R.D.; Histology, P.Z. and S.R.D.; Flow cytometry, P.Z., R.C.W., J.B.Z. and S.R.D.; TCR sequencing, R.C.W. and S.R.D.; Figure preparation, C.S., P.Z., S.G. and S.R.D.; Writing - Original Draft, S.R.D.; Writing - Review & Editing, C.S., P.Z., S.G., N.L.L.G. and S.R.D.; Supervision and Funding Acquisition, S.G, N.L.L.G. and S.R.D.

## Competing interests

S.G. and S.R.D. are inventors on a provisional patent application (2021903279) related to this research. All other authors declare no competing interests.
