## [Peer Review File · Nature Communications]

Covalent TCR-peptide-MHC interactions induce T cell activation and redirect T cell fate in the thymusREVIEWER COMMENTS

Reviewer #1 (Thymic selection, TCR signaling) (Remarks to the Author):

Szeto et al provide a compelling case that disulfide bond formation between TCR and p-MHC leads to activation of T cells, and in thymocytes leads mainly to activation-induced cell death, but also to diversion to the agonist-selected subsets of T cells.

To start, I have one minor quibble. I don't think that DSB is a standard abbreviation for disulfide bond, to me it means double strand break, which is also important in thymocyte biology. What's wrong with using "S-S bond"?

The experiments are mostly based on a TCR against a flu peptide presented on H2Db called 6218. Based on crystal structure information, they engineered a Cys residue into the TCR CDR3 α or CDR3 β (I'll refer to them as α C and β C), and made retrogenic mice. They found that few thymocytes matured through positive selection if their TCRs had the potential to make disulfide binds with peptides that presumably also had a free Cys residue available to the TCR. Most of the cells were deleted before (or at the same signaling time-point as) positive selection. The few cells from the Cys-containing retrogenic TCR that matured to TCR^{hi} stages were CD4⁻8⁻ (DN), PD-1⁺, the phenotype of precursors of the agonist-selected intra-epithelial lymphocyte population (pIEL).

Fig. 1e, showing the presence of TCR retrogenic thymocytes in relation to the cortex and medulla of the thymus, is interpreted as showing that 6218 WT cells are increased in the medulla, whereas there are few α C cells in general, and that they are not increased in the medulla compared to cortex. While I agree with the first point, I'm not so sure about the second. This figure should include some quantitation of the number of retrogenic cells in both medulla and cortex for α C and WT. TCR sequencing analysis showed that Cys in CDR3 is found reasonably often in immature thymocytes, not in mature cells, but that the cells that get through positive selection in Zap70-hypoactive Zap70^{mrt/mrd} mice don't show this pre/post selection disparity between WT, α C, and β C TCR retrogenics. Also, in MHC-null mice, the preselection thymocytes have Cys-containing CDR3s.

Overall, the interpretation that Cys-containing TCRs are generally selected against, i.e. deleted, and the few that survive are a type of self-reactive T cell that become IELs, seems sound.

Based on the crystal structure, they mutated the peptide for 6218 such that it should be able to interact with the free Cys in α C TCR. In terms of the basic on-off interaction, there was little change in the affinity compared to α C binding to the WT peptide. However, over time, a proportion of the interactions became permanent, due to formation of the covalent disulfide bond (this process inhibited by DTT). The half life of the non-covalent interaction was <1 sec, compared to >1 hour for the covalent bond. The crystal structures revealed that the covalent S-S bond showed a different rotamer of the Cys in the peptide, compared to the unbound form or to the form in complex with the WT TCR. Thus, formation of the S-S bond likely required the rotation, and thus it did not occur on each of the (short) binding events. When they compared mutations at other peptide residues on affinity, the potential for the S-S bond did not alter the affinity for the whole of the peptide.

For T cell activation in a cell line (where CD8 $\alpha\beta$ was also expressed), they found that α C was 50x more sensitive than WT TCR. This is what you would expect for a long-term assay – using IL2 secretion – rather than binding experiments. The number of fast on-off interactions will gradually change to the very slow off kind, and thus increased activation. For these T cell activation experiments, it would be good to see what are the long-term effects on things like TCR downregulation and induction of apoptosis. My guess is that although the S-S bonds lead to stronger activation in terms of IL2 secretion, they will also lead to long-term loss of TCR and to cell death.

I think that the general point of the paper is well made and clear. However, I think they could be a bit more explicit about the effect of the S-S binds on affinity. Sure, the potential for S-S bond formation doesn't increase or decrease affinity, but the fact of the bond forming in more and more

complexes over time (and maybe after many earlier on-off events) increases the affinity of the covalent linkage to approach infinity.

Minor point:

The statement “amino acids near the middle (apex) of CDR3 in TCR β (CDR3 β) often contact the peptide.” Surely this is also true for CDR3 α , not just CDR3 β ?

Reviewer #2 (T cell activation) (Remarks to the Author):

I thought this was a great paper. Multiple lines of evidence are shown to validate the fact that Cys residues in the TCR of interest allow DSBs to form between the TCR and its pMHC ligand. The data are also very convincing that these covalent interactions enhance the effects of TCR signaling and result in thymic deletion and Type A IELp formation, while non-covalent interactions induce positive selection of CD8ab+ T cells. These conclusions provide biophysical underpinning to the TCR affinity-based model of thymic selection. My only concern, and this is in the take or leave it category, is that the polyclonal T cell analysis in Fig. 2 was hard for me to follow and did not add much to the story.

Reviewer #3 (Antiviral immunity, signaling) (Remarks to the Author):

Szeto et al. demonstrate in their study the occurrence of disulfide bonds between the TCR of T cell precursors and the peptide-MHC in vivo during T cell development in the thymus. These disulfide bonds skew the development of T cells towards intestinal CD8aa T cells and away from conventional CD4+ and CD8ab T cells in TCR-retrogenic mice. They further show that disulfide bonds can overcome limited peptide affinity and thus induce functional TCR downstream signaling.

As mentioned by the authors, the fact that disulfide TCR-pMHC bonds can form and overcome low basal affinity of TCRs towards certain peptides opens new perspectives in the engineering of CAR-T cells. Furthermore, these findings would be interesting to transfer to B cells where the BCR affinity and it's resulting effect on Plasma cell or memory B cell formation has still many open questions to answer.

The authors used a broad range of techniques to analyze the biological consequences of disulfide bonds in TCR-pMHC complexes during T cell development and activation. They nicely complement each other, which makes the manuscript interesting and strengthens the message.

Please find in the following paragraphs some remarks regarding the experiments, the presentation of the results and their interpretation.

Figure 1 and 2

Some experiments have been performed with a low number of mice. Even though it is possible to perform the mentioned statistical tests, the results have low statistical power due to the low sample size. Furthermore, it is not mentioned how often these experiments have been repeated.

In Figure 1d, the number of intestinal T cells has been determined through the number of passed cells on the cytometer. Numbers should be determined by adding TrucountBeads to each sample or with similar methods to rule out bias due to time of acquisition or concentration of cells.

I would suggest to show the number of GFP+ CCR7+ or CCR7- cell in the thymus (Figure 1b), in order to strengthen the conclusion of altered T cell numbers in TCR-retrogenic mice. Differences in frequencies could arise from the change in other cell populations. Inversely, to strengthen the point that “most of the GFP+ TCRb+ thymocytes had a CD4- CD8- PD-1+ phenotype” it would be good to show the frequency of this population amongst total T cells instead of their number.

Could you please provide a gating strategy for these experiments?

Lastly, the medulla and the cortex are difficult to distinguish for 6218 TCR-retrogenic mice. Indicating the are of the medulla with a dotted line would make it easier to distinguish both regions.

Figure 6

Another point has to be made for the calculation of the EC50 in IL-2 experiments (Figure 6). It is understandable that the authors wanted to underline the differences between condition. Nonetheless, calculating the EC50 for PA4C with 6218, PA4C7L with 6218aC and PA with 6218 seems ambiguous, because the plateau of activation has not been reached.

Extended Data Figure 4

The 2Fo-Fc maps might be inversed, with blue being 3 and green being 1 sigma instead of the other way around.

Section 3 - Disulfide bond formation between TCR and peptide

The authors state that the previous results demonstrate "that a Cys-containing CDR3 elicits strong TCR signaling in vivo". This conclusion is rather correlative and could be more directly demonstrated with the EDU/HELIOS experiment in TCR-retrogenic mice, as has been done for the Zap70 mouse model.

Regarding the manuscript itself:

Some sentences and sections are incomprehensible, which is a great shame, because it prohibits the reader to fully appreciate the quality and importance of this study. Transitions between section explaining the intentions of the next step would for example simplify the reading.

Section 2

This is a particularly difficult section to follow the authors. It is rather unclear why Zap70 is of interest. Furthermore, it is not explained what the Yae62b-tg mouse model is and why it has been used. A few words on this would make it easier to follow the intentions of the experiments and the conclusions.

Section 3

The authors don't further discuss why 6218bC doesn't bind to the PA or PA4C peptides and if this result is expected.

Section 5

This section is similar to section 2 difficult to understand. Why did the authors choose P7 as an appropriate site to mutate? Furthermore, Figure 5a shows similar results for the binding of PA and the PA4C mutated peptides, but not between the 6218 and the 6218aC TCRs. The before last sentence doesn't really make sense.

Reviewer #4 (Structural biology) (Remarks to the Author):

The manuscript by Szeto et al. look to make a covalent interaction between the T cell receptor and a peptide major histocompatibility complex by a disulfide bond. A disulfide linked complex should be long lived and elicit a strong T-cell activation. The experiments appear to be well performed and the structures are well determined. My questions revolve around how to document disulfide binding between the peptide and TCR in the various experiments as many of the experiments are indirect (e.g. mutants). The crystal structure documents a disulfide bond, however crystal formation and

growth takes hours to days. For T cell activation, one would want the disulfide bond to form in a much shorter timeframe. Addition of DTT in the SPR experiment is complicated by the 6218 TCR having two disulfide bonds (from PDB ID 3PQY), which may be necessary for proper folding and function. Furthermore, the SPR methods section states that the TCR forms disulfide-linked homodimers, which required DTT to bind PA/H2-Db or PA4C/H2-Db. The oxidation state of the TCR/ PA4C/H2-Db complex needs to be verified.

#####Reviewer #1#####

Szeto et al provide a compelling case that disulfide bond formation between TCR and p-MHC leads to activation of T cells, and in thymocytes leads mainly to activation-induced cell death, but also to diversion to the agonist-selected subsets of T cells.

To start, I have one minor quibble. I don't think that DSB is a standard abbreviation for disulfide bond, to me it means double strand break, which is also important in thymocyte biology. What's wrong with using "S-S bond"?

Author response: Thank you, we agree "S-S bond" is better and have edited the manuscript accordingly.

The experiments are mostly based on a TCR against a flu peptide presented on H2Db called 6218. Based on crystal structure information, they engineered a Cys residue into the TCR CDR3 α or CDR3 β (I'll refer to them as α C and β C), and made retrogenic mice. They found that few thymocytes matured through positive selection if their TCRs had the potential to make disulfide binds with peptides that presumably also had a free Cys residue available to the TCR. Most of the cells were deleted before (or at the same signaling time-point as) positive selection. The few cells from the Cys-containing retrogenic TCR that matured to TCRhi stages were CD4 $-$ 8 $-$ (DN), PD-1+, the phenotype of precursors of the agonist-selected intra-epithelial lymphocyte population (pIEL).

Fig. 1e, showing the presence of TCR retrogenic thymocytes in relation to the cortex and medulla of the thymus, is interpreted as showing that 6218 WT cells are increased in the medulla, whereas there are few α C cells in general, and that they are not increased in the medulla compared to cortex. While I agree with the first point, I'm not so sure about the second. This figure should include some quantitation of the number of retrogenic cells in both medulla and cortex for α C and WT.

Author response: We added quantitation of the number of GFP+ cells per section to Fig. 1e. We also deleted comparisons of cortex and medulla *within* groups in order to focus on the comparisons *between* α C and WT. Thank you for this suggestion.

TCR sequencing analysis showed that Cys in CDR3 is found reasonably often in immature thymocytes, not in mature cells, but that the cells that get through positive selection in Zap70-hypoactive Zap70mrt/mrd mice don't show this pre/post selection disparity between WT, α C, and β C TCR retrogenics. Also, in MHC-null mice, the preselection thymocytes have Cys-containing CDR3s.

Overall, the interpretation that Cys-containing TCRs are generally selected against, i.e. deleted, and the few that survive are a type of self-reactive T cell that become IELs, seems sound.

Based on the crystal structure, they mutated the peptide for 6218 such that it should be able to interact with the free Cys in α C TCR. In terms of the basic on-off interaction, there was little change in the affinity compared to α C binding to the WT peptide. However, over time, a proportion of the interactions became permanent, due to formation of the covalent disulfide bond (this process inhibited by DTT). The half-life of the non-covalent interaction was <1 sec, compared to >1 hour for the covalent bond. The crystal structures revealed that the covalent S-S bond showed a different rotamer of the Cys in the peptide, compared to the

unbound form or to the form in complex with the WT TCR. Thus, formation of the S-S bond likely required the rotation, and thus it did not occur on each of the (short) binding events. When they compared mutations at other peptide residues on affinity, the potential for the S-S bond did not alter the affinity for the whole of the peptide.

For T cell activation in a cell line (where CD8 $\alpha\beta$ was also expressed), they found that αC was 50x more sensitive than WT TCR. This is what you would expect for a long-term assay – using IL2 secretion – rather than binding experiments. The number of fast on-off interactions will gradually change to the very slow off kind, and thus increased activation. For these T cell activation experiments, it would be good to see what are the long-term effects on things like TCR downregulation and induction of apoptosis. My guess is that although the S-S bonds lead to stronger activation in terms of IL2 secretion, they will also lead to long-term loss of TCR and to cell death.

Author response: In early experiments we did observe TCR downregulation, but not induction of apoptosis, in the T cell line (see figure below). However, substantial TCR β downregulation occurred upon coculture with the DC line even when peptide antigen was absent (see figure below). As IL-2 secretion was fully and consistently peptide-dependent, we focused on this readout in subsequent experiments.

Figure legend. In the coculture assay described in Figure 6, supernatant IL-2 (a) provided a more specific readout of peptide-dependent T-cell activation than TCR β downregulation (b), and there was no evidence that peptide induced apoptosis in the T-cell line (c). Data in (b) and (c) were obtained using FACS.

I think that the general point of the paper is well made and clear. However, I think they could be a bit more explicit about the effect of the S-S bonds on affinity. Sure, the “potential” for S-S bond formation doesn't increase or decrease affinity, but the fact of the bond forming in more and more complexes over time (and maybe after many earlier on-off events) increases the affinity of the covalent linkage to approach infinity.

Author response: Thank you for this suggestion. Throughout the manuscript we have now inserted the word “initial” to clarify that it is only the initial affinity of TCR association with pMHC that is not affected by the potential for S-S bond formation. In addition, the first paragraph of the Discussion now states, “after S-S bond formation, the long lifetime of the covalently bound TCR-pMHC complex mimics a very high-affinity interaction.”

Minor point:

The statement “amino acids near the middle (apex) of CDR3 in TCR β (CDR3 β) often contact the peptide.” Surely this is also true for CDR3 α , not just CDR3 β ?

Author response: We agree. We deleted “in TCR β (CDR3 β)”.

#####

#####Reviewer #2#####

I thought this was a great paper. Multiple lines of evidence are shown to validate the fact that Cys residues in the TCR of interest allow DSBs to form between the TCR and its pMHC I ligand. The data are also very convincing that these covalent interactions enhance the effects of TCR signaling and result in thymic deletion and Type A IELp formation, while non-covalent interactions induce positive selection of CD8ab+ T cells. These conclusions provide biophysical underpinning to the TCR affinity-based model of thymic selection. My only concern, and this is in the take or leave it category, is that the polyclonal T cell analysis in Fig. 2 was hard for me to follow and did not add much to the story.

Author response: We added introductory and summarizing sentences to the Results (highlighted) to clarify the aims and conclusions of the experiments described in Fig. 2.

#####

#####Reviewer #3#####

Szeto et al. demonstrate in their study the occurrence of disulfide bonds between the TCR of T cell precursors and the peptide-MHC in vivo during T cell development in the thymus. These disulfide bonds skew the development of T cells towards intestinal CD8aa T cells and away from conventional CD4+ and CD8ab T cells in TCR-retrogenic mice. They further show that disulfide bonds can overcome limited peptide affinity and thus induce functional TCR downstream signaling.

As mentioned by the authors, the fact that disulfide TCR-pMHC bonds can form and overcome low basal affinity of TCRs towards certain peptides opens new perspectives in the engineering of CAR-T cells. Furthermore, these findings would be interesting to transfer to B cells where the BCR affinity and its resulting effect on Plasma cell or memory B cell formation has still many open questions to answer.

The authors used a broad range of techniques to analyze the biological consequences of disulfide bonds in TCR-pMHC complexes during T cell development and activation. They nicely complement each other, which makes the manuscript interesting and strengthens the message.

Please find in the following paragraphs some remarks regarding the experiments, the presentation of the results and their interpretation.

Figure 1 and 2

Some experiments in have been performed with a low number of mice. Even though it is possible to perform the mentioned statistical tests, the results have low statistical power due to the low sample size. Furthermore, it is not mentioned how often these experiments have been repeated.

Author response: In Extended Data Fig. 1, we now provide data from a separate TCR-retrogenic experiment using *Tcra*^{-/-} BM donors and recipients, producing similar results.

In Figure 1d, the number of intestinal T cells has been determined through the number of passed cells on the cytometer. Numbers should be determined by adding TrucountBeads to each sample or with similar methods to rule out bias due to time of acquisition or concentration of cells.

Author response: We agree. To exclude those potential confounders, Fig. 1d now reports the percentage of intestinal T cells among CD45+ cells.

I would suggest to show the number of GFP+ CCR7+ or CCR7- cell in the thymus (Figure 1b), in order to strengthen the conclusion of altered T cell numbers in TCR-retrogenic mice. Differences in frequencies could arise from the change in other cell populations. Inversely, to strengthen the point that “most of the GFP+ TCRb+ thymocytes had a CD4- CD8- PD-1+ phenotype” it would be good to show the frequency of this population amongst total T cells instead of their number.

Author response: We agree, thank you. These changes have been made to Fig. 1b and the text revised accordingly.

Could you please provide a gating strategy for these experiments?

Author response: Gating strategies for the TCR-retrogenic analyses are now provided in Extended Data Fig. 1.

Lastly, the medulla and the cortex are difficult to distinguish for 6218 TCR-retrogenic mice. Indicating the area of the medulla with a dotted line would make it easier to distinguish both regions.

Author response: This has been done in the revised manuscript, thank you.

Figure 6

Another point has to be made for the calculation of the EC50 in IL-2 experiments (Figure 6). It is understandable that the authors wanted to underline the differences between condition. Nonetheless, calculating the EC50 for PA4C with 6218, PA4C7L with 6218aC and PA with 6218 seems ambiguous, because the plateau of activation has not been reached.

Author response: We revised the figure legend to say, “For curves that did not reach a plateau, the reported EC₅₀ values provide a minimum estimate of the EC₅₀”.

Extended Data Figure 4

The 2Fo-Fc maps might be inverted, with blue being 3 and green being 1 sigma instead of the other way around.

Author response: We have checked the figure and there is no mistake but to clarify, the blue and green maps are different, with blue being 2Fo-Fc and contoured at 1 sigma showing the density after refinement, while the green maps show unbiased Fo-Fc contoured at 3 sigma and is showing the clear density for the peptide after molecular replacement and before building the peptide in the structure.

Section 3 - Disulfide bond formation between TCR and peptide

The authors state that the previous results demonstrate "that a Cys-containing CDR3 elicits strong TCR signaling in vivo". This conclusion is rather correlative and could be more directly demonstrated with the EDU/HELIOS experiment in TCR-retrogenic mice, as has been done for the Zap70 mouse model.

Author response: We feel the revised manuscript better describes several types of evidence, based on mice with attenuated Zap70 function, "that a Cys-containing CDR3 elicits strong TCR signaling in vivo". EDU/HELIOS analysis in TCR-retrogenics is attractive but is currently beyond our capabilities as we do not have access to *Rag*^{-/-} mice with defective apoptosis and we do not have a method to co-stain for GFP and EDU/HELIOS in FACS.

Regarding the manuscript itself:

Some sentences and sections are incomprehensible, which is a great shame, because it prohibits the reader to fully appreciate the quality and importance of this study. Transitions between section explaining the intentions of the next step would for example simplify the reading.

Author response: Thank you for raising this concern. We added introductory and summarizing sentences to the text (highlighted) and feel the manuscript is much improved.

Section 2

This is a particularly difficult section to follow the authors. It is rather unclear why Zap70 is of interest. Furthermore, it is not explained what the Yae62b-tg mouse model is and why it has been used. A few words on this would make it easier to follow the intentions of the experiments and the conclusions.

Author response: This has been addressed in our revised manuscript.

Section 3

The authors don't further discuss why 6218bC doesn't bind to the PA or PA4C peptides and if this result is expected.

Author response: The revised manuscript explains that this result was expected and why.

Section 5

This section is similar to section 2 difficult to understand. Why did the authors choose P7 as an appropriate site to mutate?

Author response: The revised manuscript explains that P7 is known to be critical for TCR recognition of this ligand, hence we expected P7 substitutions would result in lower-affinity TCR-pMHC interactions.

Furthermore, Figure 5a shows similar results for the binding of PA and the PA4C mutated peptides, but not between the 6218 and the 6218aC TCRs. The before last sentence doesn't really make sense.

Author response: The sentence in question has been deleted.

#####

#####Reviewer #4 (Structural biology) #####

The manuscript by Szeto et al. look to make a covalent interaction between the T cell receptor and a peptide major histocompatibility complex by a disulfide bond. A disulfide linked complex should be long lived and elicit a strong T-cell activation. The experiments appear to be well performed and the structures are well determined. My questions revolve around how to document disulfide binding between the peptide and TCR in the various experiments as many of the experiments are indirect (e.g. mutants). The crystal structure documents a disulfide bond, however crystal formation and growth takes hours to days. For T cell activation, one would want the disulfide bond to form in a much shorter timeframe. Addition of DTT in the SPR experiment is complicated by the 6218 TCR having two disulfide bonds (from PDB ID 3PQY), which may be necessary for proper folding and function. Furthermore, the SPR methods section states that the TCR forms disulfide-linked homodimers, which required DTT to bind PA/H2-Db or PA4C/H2-Db. The oxidation state of the TCR/PA4C/H2-Db complex needs to be verified.

Author response: We understand the main issue here concerns the effects of DTT in the SPR experiments.

Initially, the reviewer seems concerned that the DTT could cause protein unfolding. Protein unfolding may explain the effect in Figure 4b, where the presence of DTT in the pMHC analyte abrogates the long-lived subpopulation of TCR-pMHC complexes. However, data in Figure 4a-c demonstrate the noncovalent 6218 TCR-PA4C/H2-D^b interaction was indistinguishable in the presence or absence of 2 mM DTT in the pMHC analyte, a result that excludes the hypothesis that this treatment causes TCR or pMHC unfolding. Indeed, one of us has demonstrated pMHC molecules remain intact after treatment with 10 mM DTT (doi.org/10.4049/jimmunol.1401357).

Then, the reviewer raises a concern about the DTT treatment of the immobilized 6218 α C TCR. As above, the reviewer may be concerned that this treatment causes unfolding of the TCR, followed by refolding, potentially exposing cysteines outside the CDR3 for the formation of S-S bonds with the PA4C/H2-D^b. However, the DTT-treated 6218 α C TCR can bind to PA/H2-D^b, a result that excludes the hypothesis that the 6218 α C TCR has refolded into a non-native conformation.

Another possibility is that the reviewer is concerned that the effect of DTT on the immobilized 6218 α C TCR is long-lasting. If this were the case, then the DTT might antagonize S-S bond formation between the 6218 α C TCR and the PA4C peptide. We can exclude this possibility because 1-minute injections of pMHC produced different results from injections that lasted 5, 20, or 50 minutes. If it existed, the inhibitory effect of DTT should have been consistent because the pMHC injection durations are short compared to the 3-hour equilibration phase that followed DTT treatment of the immobilized 6218 α C TCR. According to Sigma-Aldrich's product information, DTT has a half-life of 1.4 hours at the pH and temperature conditions of the SPR experiments.

The finding that a long-lived subpopulation of TCR-pMHC complexes was detectable after 1 minute of pMHC injection, coupled with the observation that long-lived TCR-pMHC complexes increased at the expense of short-lived TCR-pMHC complexes during injections that lasted 5, 20, or 50 minutes (Figure 4d and compare Figure 5d with Extended Data Figure

6c) demonstrates that new S-S bonds formed within a timeframe of minutes in the SPR experiments. As stated by the reviewer, T-cell activation elicited by a covalent TCR-peptide bond likely requires S-S bond formation within a timeframe of minutes, as observed in the SPR data.

To clarify, we added data in Extended Data Figure 6 to demonstrate the immobilized 6218 α C TCR without DTT treatment fails to bind to pMHC ligands (panel **a**). However, a 10-minute treatment of the immobilized 6218 α C TCR with 1 mM DTT, followed by 3 hours of running buffer injection without DTT, enables the immobilized 6218 α C TCR to bind to pMHC ligands in SPR experiments (panel **b**). Panel **b** also illustrates how we used the extent of binding to PA/H2-D^b to account for differences between biosensor chips in the amount of immobilized 6218 α C TCR available for binding. These new data panels make explicit our approach to enable and quantify pMHC binding to the immobilized 6218 α C TCR in the SPR experiments.

The SPR results are consistent with and complement the X-ray crystallography and tetramer binding and dissociation data. Together, these data directly document disulfide binding between the peptide and TCR.

REVIEWERS' COMMENTS

Reviewer #1 (Remarks to the Author):

They have responded well to all of my points, and, as far as I can see, to those of the other reviewers. I think this is a fine paper.

Reviewer #3 (Remarks to the Author):

I thank the authors for their answers and for taking into account my suggestions regarding the manuscript. It is now much easier to follow. Adding the gating strategy and some supplementary information makes the study more robust.

Reviewer #4 (Remarks to the Author):

The authors have addressed my concerns with additional data, literature citations and a comprehensive discussion in the rebuttal. I support publication of the manuscript.